# The weekend effect in kidney transplantation outcomes: A meta-analysis

**Haifeng Wang**[1], **Yi Yi**[1], **Tan Xiao**[1], **Aiqing Li**[1], **Yongfei Liu**[1], **Xiaoli Huang**[2]*

**1** Department of Urology, Longyan First Affiliated Hospital of Fujian Medical University, Longyan, Fujian, China, **2** Department of Hematology, Longyan First Affiliated Hospital of Fujian Medical University, Longyan, Fujian, China

* 973229194@qq.com

**Data Availability Statement:** All relevant data are within the paper and its Supporting Information files.

**Funding:** The authors received no specific funding for this work.

## Abstract

### Objective

To determine whether kidney transplants performed during weekends have worse outcomes than those performed during weekdays.

### Methods

For this systematic review, PubMed, EMBASE, and the Cochrane Library (January 2000 to January 2023) were searched. We examined the survival rates of patients and grafts for hospital inpatients admitted during weekends and those admitted during weekdays. To be included, the study had to be in English and had to provide discrete survival data around weekends versus weekdays, including patients who were admitted as inpatients over the weekend.

### Results

Five studies (n = 163,506 patients) were examined. The hazards ratio (HR) of the survival rate of patients with weekend transplantation was 1.01 (95% confidence interval [CI], 0.96 to 1.06) when compared with patients with weekday transplantation. Patients who had renal transplant on weekends had an overall allograft survival HR of 1.01 (95% CI, 0.99 to 1.03) and death-censored allograft survival HR of 1.01 (95% CI, 0.98 to 1.04). Comparison of length of hospital stay, rejection, surgical complications, and vascular complications between renal transplants on weekends and those on weekdays showed no statistical difference.

### Conclusion

Hospital inpatients admitted for renal transplantation during weekends have a survival rate similar to that of inpatients admitted during weekdays. The weekend effect of renal transplantation was very weak; hence, transplantations done during weekends and weekdays are both appropriate.

**Competing interests:** The authors have declared that no competing interests exist.

## 1. Introduction

The "weekend effect" is the finding of worse outcomes in patients admitted on the weekend than in those admitted during weekday and that patients admitted on the weekend had higher rates of adverse events and death [1]. Over the past several decades, the issue of the weekend effect has been reported by several studies [2–4], which have suggested an increasing global mortality in patients admitted to hospitals during weekends compared with those admitted during weekdays. Many studies [5–10] have shown a significantly worse outcome among patients admitted on a weekend for various emergency medical and surgical conditions, including myocardial infarction, aortic dissection, pulmonary embolism, or stroke, and many other critical illnesses requiring admissions in the intensive care units. Al Zamel et al. [11] pointed out that the reduction of experienced staff during weekends may lead to insufficient nursing care, which, in turn, may lead to aggravation or irreversibility of illness. Groves et al. [12] believe that patients admitted to hospitals on weekends may be unwilling to undergo surgery or experience delay in receiving necessary intervention, which leads to poor prognosis. Another possibility is that there may be differences in the severity of diseases or complications between patients admitted on weekends and those admitted on weekdays [13]. Some studies [14] have reported a difference in graft survival rate between patients admitted in hospitals on weekends and weekdays. However, whether there is a weekend effect in patients who undergo renal transplantation is uncertain. Because of the inconsistency between studies on the existence of this effect, we aimed to determine, through a meta-analysis, whether weekend transplantation had any influence on graft and patient survival rates. In this study, we also aimed to examine the association between the day (weekend versus weekday) of kidney transplant surgery on both complications and rejection.

## 2. Methods

### 2.1. Data sources and searches

We performed a literature search (January 2000 to January 2023) of multiple databases, including PubMed, EMBASE, and the Cochrane Library. Two reviewers (Haifeng Wang and Yi Yi) independently evaluated the full article of each abstract. Any dispute was resolved by a third reviewer (Xiaoli Huang). The terms "weekend," "weekend effect," and "renal or kidney transplantation" were searched in the title and abstract. The related-articles function was used to broaden the search, all conference abstracts, and retrieved studies.

### 2.2. Study selection

To be included in the systematic review, the study had to be in English and had to provide mortality data on weekends (including holidays) versus weekdays, including patients who were admitted as inpatients over the weekend. We excluded studies that were letters, editorials, case reports, and so on.

### 2.3. Data extraction

Three experienced investigators (Tan Xiao, Aiqing Li and Yongfei Liu) independently analyzed the final defined articles for the primary and secondary parameters. The primary parameters included patient, overall, and death-censored graft survival, and secondary parameters included the initial length of hospital stay after transplantation, delayed allograft function (defined as use of dialysis in the first week after transplantation), and acute rejection (if treated for acute rejection, with or without biopsy) within the first year after transplantation. Unexpected discrepancies were carefully discussed and resolved during this process.

## 2.4. Quality assessment

Standard quality evaluation of the five included studies [14–18] was performed using the New-castle-Ottawa Quality Assessment Scale.

## 2.5. Data synthesis and analysis

Relevant parameters explored using RevMan software (Version 5.3; Cochrane Collaboration, Oxford, UK) included operative time, warm ischemia time, estimated blood loss, and length of hospital stay. For continuous data, we calculated the weighted mean difference with 95% confidence intervals (CIs). For dichotomous data, odds ratios (ORs) were calculated with 95% CI. Heterogeneity among studies was measured using the $I^2$ and Cochran's Q tests (Mantel-Haenszel $\chi^2$ test). In cases of significant heterogeneity ($\chi^2$ $P$ value $\geq 0.1$), a fixed-effect model was selected. Otherwise, a random-effects model was used.

# 3. Results

## 3.1. Literature selection and screening

A literature search retrieved 987 unique citations, of which 187 studies were obtained from PubMed, 456 articles were from EMBASE, and 344 articles were from the Cochrane Library. Thirty-seven publications were subsequently eliminated because they were obvious duplicates, and another 570 were excluded after title and abstract review, which included irrelevant research studies, reviews, and case reports. Three hundred eighty publications were full-text articles, of which only five articles (n = 163,506 patients) met the criteria mentioned above and were recommended for this meta-analysis. Across all five articles, 28% and 72% of the patients underwent transplantation on a weekday and a weekend, respectively. The study selection and screening process is shown in Fig 1.

## 3.2. Characteristics of the included studies

The articles were published between 2001 and 2021. Table 1 presents a summary of the study characteristics and outcomes of accepted articles.

## 3.3. Quality assessment of the included studies

Standard quality evaluation based on the Newcastle-Ottawa Quality Assessment Scale showed that the five included studies were reliable (Tables 1 and 2). However, the method used to select the specimens may have contributed to bias.

## 3.4. Pooled results

**3.4.1. Primary outcomes.** *3.4.1.1. Patient survival.* Four studies [14,16–18] contributed to the unadjusted analysis of patient survival (Fig 2A). Patient survival was defined as the time from kidney transplant to death (from any cause) or last contact with the patient alive. The pooled studies were homogenous ($P$ = 0.78). With the fixed-effects model, the HR of the survival rate of patients with weekend transplantation was 1.01 (95% CI, 0.96 to 1.06) when compared with patients with weekday transplantation. However, the pooled results revealed no significant difference between the weekend and weekday groups in terms of patient survival rate ($P$ = 0.78).

*3.4.1.2. Overall allograft survival.* Overall, five studies (n = 163,506 patients) [14–18] assessed the association between weekend transplantation and its effects on patient allograft survival. Overall graft survival was defined as the time from kidney transplant to death (from

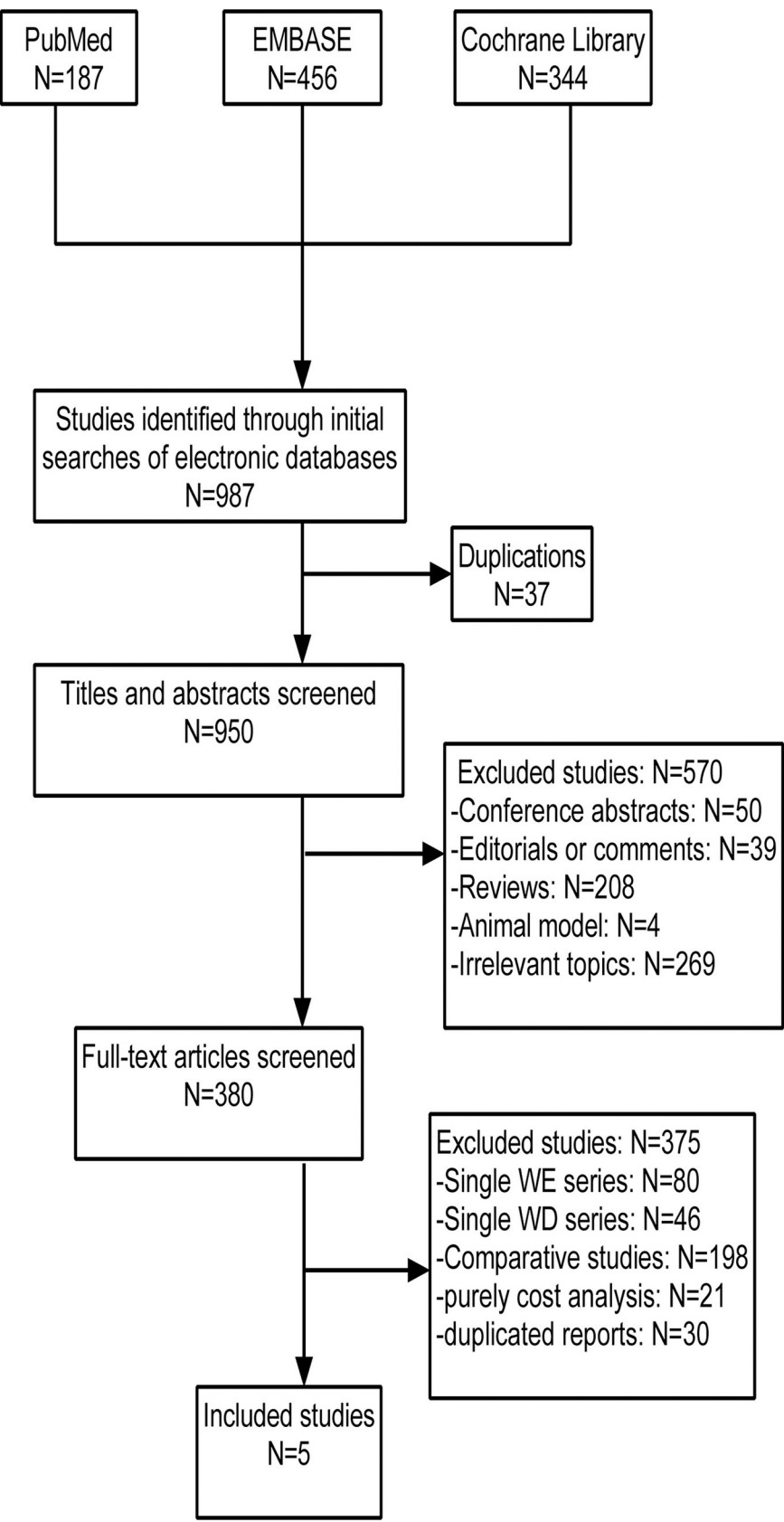

**Fig 1. Flow diagram of the studies identified, included, and excluded.**

**Table 1. Characteristics of included studies.**

| Author(s) | Year | Country | Design type | No. of patients | | Follow-up, mo | Quality score |
|---|---|---|---|---|---|---|---|
| | | | | Weekend | Weekday | | |
| Baid-Agrawal, S.et al | 2016 | USA | R | 37,654 | 99,061 | 54.6 | 7 |
| Lim, W.H. et al | 2019 | Australia, New Zealand | R | 1,868 | 4,754 | N | 7 |
| Schutte-Nutgen, K.et al | 2017 | Germany | R | 164 | 461 | N | 5 |
| Ville, S. et al | 2020 | France | R | 1,999 | 4,643 | 48 | 6 |
| Anderson, B.M.et al | 2017 | UK | C | 5,178 | 7,724 | N | 7 |

R, retrospective study; C, cohort study; N, none; mo, months.

any cause), graft failure, or last contact, whichever occurred first. There was no associated heterogeneity ($P = 0.51$). Compared with transplant recipients on weekdays, transplant recipients on weekends had an HR of overall allograft survival of 1.01 (95% CI, 0.99 to 1.03). However, there was no significant difference in the items between the two groups ($P = 0.50$) (Fig 2B).

*3.4.1.3. Death-censored allograft survival.* Death-censored allograft survival, defined as the time from kidney transplant to graft failure, has been reported in two studies [14,16]. There was no associated heterogeneity ($P = 0.85$). Furthermore, there was no significant difference in the risk of overall allograft survival between the weekend and weekday groups (HR, 1.01; 95% CI, 0.98 to 1.04; $P = 0.45$ (Fig 2C).

**3.4.2. Secondary outcomes.** The length of hospital stay was measured in two studies [14,18]. A meta-analysis of these studies showed no significant difference between the weekend and weekday groups (mean difference (MD), -0.50; 95% CI, −1.48 to 0.48; $P = 0.32$) (Fig 2D). The pooled data [14–16] showed no significant statistical difference in rejection between the two groups (OR, 1.00; 95% CI, 0.97 to 1.04; $P = 0.94$) (Fig 2E). In two studies [16,17], surgical complications including bleeding and urinary fistula, have been reported. There was no significant difference between the two groups (OR, 0.98; 95% CI, 0.59 to 1.62; $P = 0.92$) (Fig 2F). Vascular complications included graft stenosis and thrombosis. No statistical difference was found between the two groups in the two studies (15–17) (OR, 1.19; 95% CI, 0.75 to 1.89; $P = 0.45$) (Fig 2G) Table 3.

## 3.5. Sensitivity analysis

A sensitivity analysis was performed by repeating the analysis after sequential exclusion of one study at a time and using fixed- and random-effect models, respectively, to observe the effect on the whole results in our meta-analysis, and the results were consistent.

## 3.6. Assessment of publication bias

A funnel diagram was drawn to investigate the presence of publication bias in this study. The funnel diagram is symmetrical, indicating no publication bias in the literature of this study (Figs 3 and S1).

**Table 2. Quality evaluation of included studies based on the newcastle-ottawa quality assessment scale.**

| Author(s) | Year | Selection | Comparability | Outcome | Total score of NOS evaluation |
|---|---|---|---|---|---|
| Baid-Agrawal, S.et al | 2016 | ☆☆☆ | ☆☆ | ☆☆ | 7 |
| Lim, W.H. et al | 2019 | ☆☆☆ | ☆☆ | ☆☆ | 7 |
| Schutte-Nutgen, K.et al | 2017 | ☆☆ | ☆☆ | ☆ | 5 |
| Ville, S. et al | 2020 | ☆☆ | ☆☆ | ☆☆ | 6 |
| Anderson, B.M.et al | 2017 | ☆☆☆ | ☆☆ | ☆☆ | 7 |

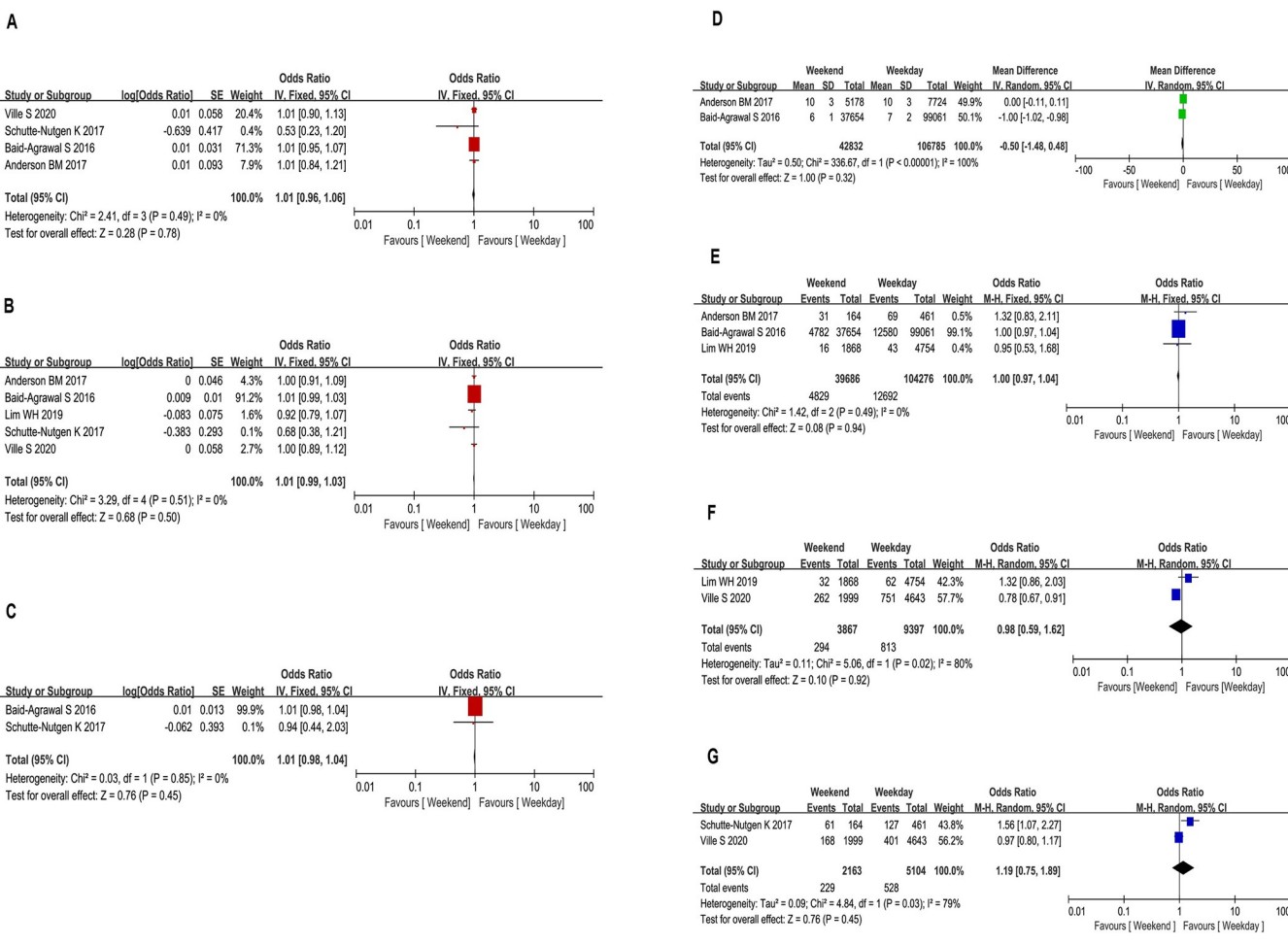

**Fig 2.** Forest plot and meta-analysis comparing weekend and weekday transplantation in terms of patient survival (A), overall allograft survival (B), death-censored allograft survival (C), length of hospital stay (D), rejection (E), surgical complications (F), and vascular complications (G).

## 4. Discussion

This meta-analysis examined five retrospective studies involving 163,506 patients and compared the clinical outcomes between the weekend and weekday groups to determine whether kidney transplants performed during weekends have worse outcomes than those performed during weekdays. Results showed no significant differences in patient and graft survival. Most deceased-donor kidney transplants are emergency operations, and the time point of transplantation depends on the time point of kidney acquisition. The cold ischemia time of most deceased-donor kidneys is around 5–10 hours, and a longer cold ischemia time can lead to an increase in the proportion of delayed renal function recovery after transplantation [19,20].

Our data also conflict with published observations on the weekend effect that would be considered relevant to deceased-donor kidney allograft recipients. By integrating the latest research results, our study found no difference in survival between patients and allografts. As all transplantation-related standardized procedures, including matching, allocation, scores promoting special subgroups of patients, immunosuppression protocols and early post-op care are standardized for years. Furthermore, a well-trained and experienced transplant surgeon can greatly promote the effectiveness of transplantation. The decision-making algorithm during the transplantation process is clear, and low-level physicians always consult with senior

**Table 3. Results of meta-analytic comparison of weekend and weekday transplantation.**

| Outcomes of interest | Studies, no. | Weekend patients, no. | Weekday patients, no. | WMD/OR (95% CI) | p value* | Study heterogeneity | | | |
|---|---|---|---|---|---|---|---|---|---|
| | | | | | | $\chi^2$ | df | $I^2$, % | P value* |
| **Primary outcomes** | | | | | | | | | |
| Patient survival | 4 | 44,995 | 111,889 | 1.01 [0.90, 1.13] | 0.78 | 2.41 | 3 | 59 | 0.49 |
| Overall allograft survival | 5 | 46,863 | 116,643 | 1.01 [0.99, 1.03] | 0.50 | 3.29 | 4 | 0 | 0.51 |
| Death-censored allograft survival | 2 | 37,818 | 99,522 | 1.01 [0.98, 1.04] | 0.45 | 0.03 | 1 | 0 | 0.85 |
| **Secondary outcomes** | | | | | | | | | |
| The length of hospital stay, d | 2 | 42,832 | 106,785 | -0.50 [-1.48, 0.48] | 0.32 | 336.67 | 1 | 100 | **<0.00001** |
| Rejection | 3 | 39,686 | 104,276 | 1.00 [0.97, 1.04] | 0.94 | 1.42 | 2 | 0 | 0.49 |
| Surgical complications | 2 | 3,867 | 9,397 | 0.98 [0.59, 1.62] | 0.92 | 5.06 | 1 | 80 | **0.02** |
| Vascular complications | 2 | 2,163 | 5,104 | 1.19 [0.75, 1.89] | 0.45 | 4.84 | 1 | 79 | **0.03** |

* Statistically significant results are shown in bold.

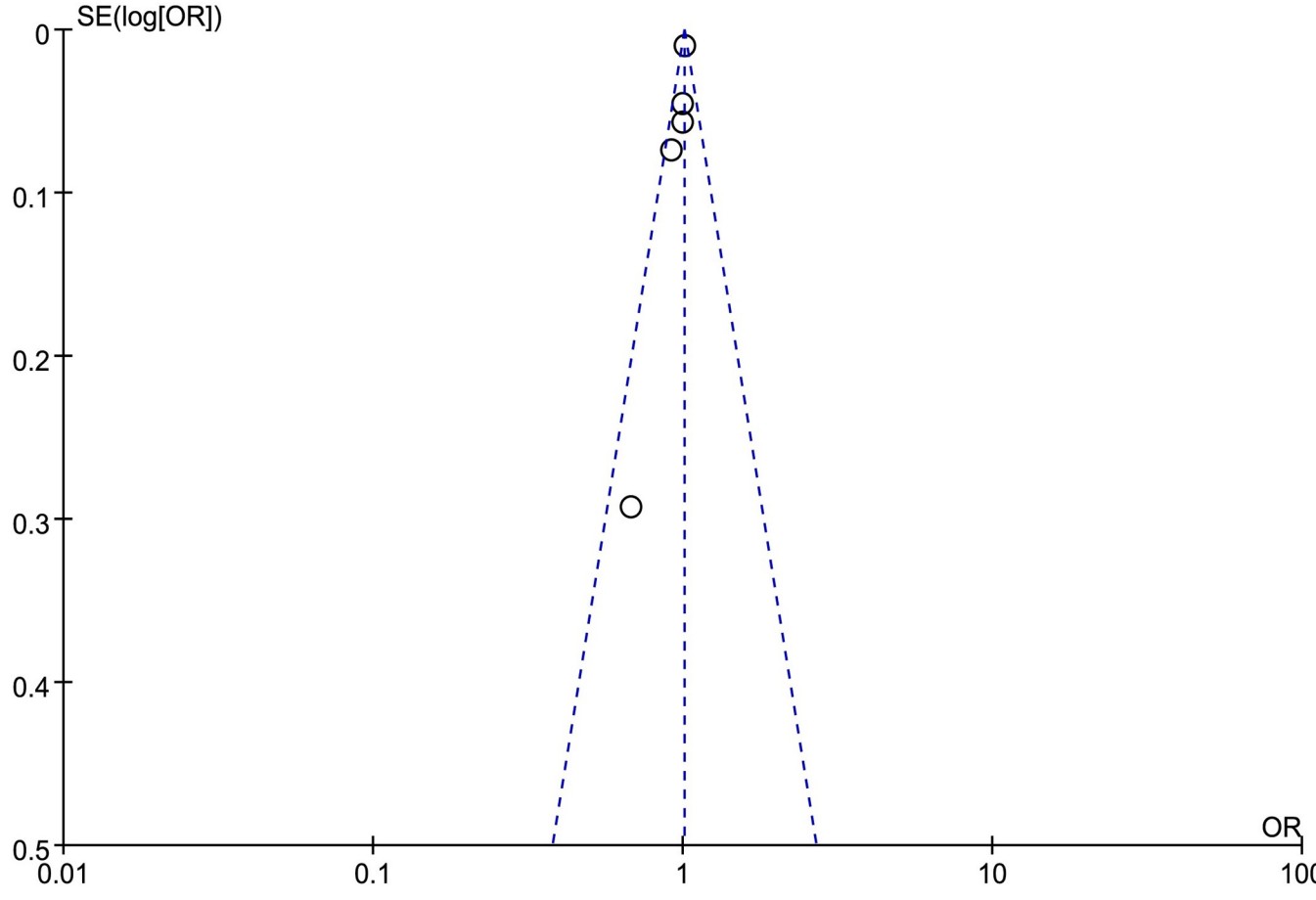

**Fig 3. Funnel plot of overall allograft survival.**

attending physicians over the phone (if necessary) in individual cases of problems or any uncertainty. When the kidney transplant is performed in the middle of the night, the surgeon may not be able to concentrate due to fatigue, which may be related to the better effect of the operation at the beginning of the day on weekends, but the overall results of transplantation on weekends and weekdays are the same [21].

Early allograft failures attributed to vascular complications were more common in weekend transplants than in weekday transplants. Most failures caused by vascular complications occur within the first 7 days post-transplant, leading to short-term transplant failure and long-term reduced patient and graft survival rates. One reason is that the majority of post-kidney transplantation care is ward-based but remains predominantly led by senior medical or surgical reviews all days of the week, possibly shielding kidney allograft recipients from any weekend effect [22]. In addition, there are the most experienced transplant team members who can access vascular imaging services in a timely manner, especially for patients with a large burden of vascular diseases involved in complex vascular/anastomotic transplantation. Nevertheless there are different service systems in different transplant centers, always there is some system of duty-on-call, allowing to coordinate the presence of licensed transplant surgeon during week-ends, to perform the procedure, effectively avoiding the occurrence of transplant complications. Transplantation doctors undergo specialized training and are more alert than regular surgeons. Therefore, they are very adept with technology and and treatment of complications, and there is little impact between the two groups in this regard.

However, our study also has limitations. First, only a few studies were included, and the limited sample size may have affected the results of the included studies. Second, confounding factors, such as differences in research time, technological innovation, and the contribution of resource limitations, have impact on research [23,24]. Finally, the meta-analysis should be tested for a heterogeneity. If heterogeneity exists, it is best to analyze the causes of heterogeneity. We will follow up more relevant research, update data in time and report the latest research results.

Nevertheless, our study is meaningful. The results of a single-center study are independent. Meta-analysis combines the effects of these studies, which can better evaluate the impact of the two groups on renal transplantation.

In conclusion, the weekend effect of renal transplantation is very weak; thus, weekend and weekday transplantations are both appropriate.

## Supporting information

**S1 Fig. Funnel plot of patient survival.**
(TIF)

**S1 Table. PRISMA 2020 main checklist and PRIMSA abstract checklist.**
(DOCX)

## Author Contributions

**Conceptualization:** Xiaoli Huang.

**Data curation:** Tan Xiao, Yongfei Liu.

**Methodology:** Yi Yi, Aiqing Li.

**Resources:** Tan Xiao.

**Software:** Yi Yi, Aiqing Li.

**Supervision:** Xiaoli Huang.

**Validation:** Yongfei Liu.

**Writing – original draft:** Haifeng Wang.

**Writing – review & editing:** Haifeng Wang, Xiaoli Huang.

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
