## [Decision Letter · Decision Letter 0]

5 Apr 2023

PONE-D-23-05474The weekend effect in kidney transplantation outcomes: a meta-analysisPLOS ONE

Dear Dr. Huang,

Thank you for submitting your manuscript to PLOS ONE. After careful consideration, we feel that it has merit but does not fully meet PLOS ONE’s publication criteria as it currently stands. Therefore, we invite you to submit a revised version of the manuscript that addresses the points raised during the review process.

Please revise.

We look forward to receiving your revised manuscript.

Kind regards,

Academic Editor

PLOS ONE

Journal Requirements:

- https://doi.org/10.1371/journal.pone.0190227

- 10.1097/TP.0000000000001522

- https://doi.org/10.1111/tri.13377

In your revision ensure you cite all your sources (including your own works), and quote or rephrase any duplicated text outside the methods section. Further consideration is dependent on these concerns being addressed.

Reviewers' comments:

Reviewer's Responses to Questions

**Comments to the Author**

1. Is the manuscript technically sound, and do the data support the conclusions?

Reviewer #1: Partly

Reviewer #2: Yes

Reviewer #3: Yes

2. Has the statistical analysis been performed appropriately and rigorously? 

Reviewer #1: Yes

Reviewer #2: I Don't Know

Reviewer #3: Yes

3. Have the authors made all data underlying the findings in their manuscript fully available?

Reviewer #1: Yes

Reviewer #2: Yes

Reviewer #3: Yes

4. Is the manuscript presented in an intelligible fashion and written in standard English?

Reviewer #1: No

Reviewer #2: Yes

Reviewer #3: Yes

5. Review Comments to the Author

Reviewer #1: The idea of this research was to define, whether there is any impact of week-end related kidney transplantation on the long-term outcomes. Some studies, in which the information of the timing of surgery was included – have been used for meta-analysis. The final conclusion is that there is no difference in graft survival, nevertheless when a procedure was performed. The data used for meta-analysis come from different European countries, USA and Australia-Zew Zealand.

The final result is not surprising, as all transplantation-related procedures, including matching, allocation, scores promoting special subgroups of patients, immunosuppression protocols, early post-op care - are standardized for years and nevertheless some minor site or country) - related disparities, are the same on week-end and during other days. The decision making algorithms are clear and are always consulted by phone (if necessary) with senior attending physicians in individual cases of trouble or any uncertainty.

The only factor which might be of interest is the experience of transplant surgeons, operating during week-ends, in a surgical technique of kidney transplantation and impact of this on the early surgical complications (and in longer perspective) – on graft outcome (driven by surgical problems). Again, nevertheless there are different systems of being on service in different transplant centers, always there is some system of duty-on-call, allowing to coordinate the presence of licensed transplant surgeon during week-ends, to perform the procedure.

This meta-analysis, as a medical research manuscript - is simplified, and the discussion touches just a top of a hidden iceberg. The quality therefore is limited, nevertheless a statistical/methodological workload, partially provided by a contemporary artificial intelligence. The discussion must be extended, describe at least the meaning of early surgical complication on early vs late graft function and survival in details. I am not sure, the authors are the transplant physicians and whether the feel all specific clinical aspects of this problem,

Reviewer #2: This is a well written meta-analysis which answers a relevant clinical question and the results is reassuring for patients and physicians.

Very well written introduction, and reasonable discussion. Congratulations to Authors!

My point which you may use in this manuscript is about the limitation of access to the OR in some health care systems. This issue usually ends to starting the harvest after regular OR time during the day to be able to accommodate high surgical volume. This usually results in starting the recipient after midnight. Despite that in some health care systems, there might be an increase in surgical fee if the case starts after midnight, the surgeons are really tired at the time of the surgery. This may have something to do with the better results on weekend when the surgery starts during the day time and may be the reason for having overall the same outcome.

You are welcome to use my point.

Reviewer #3: The manuscript is now much improved. Only suggestion is that in the discussion the authors mention that kidney transplantation is an emergency operation which is not factually correct since one can survive on dialysis. I think what the authors are trying to say is that there is a narrow window of time between organ harvesting to organ transplantation to maintain organ viability.

6. PLOS authors have the option to publish the peer review history of their article (what does this mean?). If published, this will include your full peer review and any attached files.

Reviewer #1: No

Reviewer #2: **Yes: **Hamidreza Abdi

Reviewer #3: **Yes: **Anirban Ganguli

---

## [Author Response · Author response to Decision Letter 0]

13 May 2023

1.Response to Reviewer #1 :

We appreciate for your warm work earnestly, and hope that the correction will meet with approval. We explored that there was no difference in survival rates between the graft and the recipient, and our discussion of standardized procedures and decision algorithms for the transplant process was clear. In addition, we discussed the experience of transplant surgeons during weekend surgeries and their impact on early surgical complications and transplant outcomes. Similarly, although there are different service systems in different transplant centers, always there is some system of duty-on-call, allowing to coordinate the presence of licensed transplant surgeon during week-ends, to perform the procedure, effectively avoiding the occurrence of transplant complications. We apologize that the quality of our research is limited, but our research is meaningful. We will further update the latest research, although the statistical methods used in the study can be replaced by artificial intelligence. In addition, we further explored the significance of early surgical complications on early and late kidney transplant function and survival.

2.Response to Reviewer #2 :

Thank you very much for appreciating my article. The literature for meta-analysis comes from databases available online. If access to some restricted databases is available, the number of surgeries included in the literature will increase. Of course, better results may be possible, and we will also update the evaluation in the future. Secondly, we also added in the discussion that "if a kidney transplant is performed in the middle of the night, the surgeon may not be able to concentrate due to fatigue, which may be related to the better effect of the surgery at the beginning of the day on weekends, but the overall results of the transplant on weekends and weekdays are the same." We fully agree with this valuable opinion.

3.Response to Reviewer #3 :

Most deceased-donor kidney transplants are emergency operations, and the time point of transplantation depends on the time point of kidney acquisition. The cold ischemia time of most deceased-donor kidneys is around 5-10 hours, and a longer cold ischemia time can lead to an increase in the proportion of delayed renal function recovery after transplantation.

---

## [Decision Letter · Decision Letter 1]

5 Jun 2023

The weekend effect in kidney transplantation outcomes: a meta-analysis

PONE-D-23-05474R1

Dear Dr. Huang,

We’re pleased to inform you that your manuscript has been judged scientifically suitable for publication and will be formally accepted for publication once it meets all outstanding technical requirements.

Kind regards,

Academic Editor

PLOS ONE

Additional Editor Comments (optional):

Reviewers' comments:

Reviewer's Responses to Questions

**Comments to the Author**

1. If the authors have adequately addressed your comments raised in a previous round of review and you feel that this manuscript is now acceptable for publication, you may indicate that here to bypass the “Comments to the Author” section, enter your conflict of interest statement in the “Confidential to Editor” section, and submit your "Accept" recommendation.

Reviewer #2: All comments have been addressed

Reviewer #3: All comments have been addressed

2. Is the manuscript technically sound, and do the data support the conclusions?

Reviewer #2: Yes

Reviewer #3: Yes

3. Has the statistical analysis been performed appropriately and rigorously? 

Reviewer #2: I Don't Know

Reviewer #3: Yes

4. Have the authors made all data underlying the findings in their manuscript fully available?

Reviewer #2: Yes

Reviewer #3: Yes

5. Is the manuscript presented in an intelligible fashion and written in standard English?

Reviewer #2: Yes

Reviewer #3: Yes

6. Review Comments to the Author

Reviewer #2: The revision looks better. There are many confounding factors for making a conclusion based on this work, however, this may be considered as a signal to do more work in future.

Reviewer #3: I would appreciate if-as a response to the query made not only by me or the other reviewers who perused the manuscript-the authors make it explicitly clear if the changes suggested have been incorporated. As I compare the original and the revised manuscript, the authors did change the wordings, copied and pasted the sentences into the response to reviewer without acknowledging the same.

7. PLOS authors have the option to publish the peer review history of their article (what does this mean?). If published, this will include your full peer review and any attached files.

Reviewer #2: **Yes: **Hamidreza Abdi

Reviewer #3: No

---

## [Editor Report · Acceptance letter]

8 Jun 2023

PONE-D-23-05474R1 

*The weekend effect in kidney transplantation outcomes: a
meta-analysis*

Dear Dr. Huang:

I'm pleased to inform you that your manuscript has been deemed suitable for publication in PLOS ONE. Congratulations! Your manuscript is now with our production department. 

Kind regards, 

on behalf of

Dr. Robert Jeenchen Chen 

Academic Editor

PLOS ONE